# A novel histopathological classification of implant periapical lesion: A systematic review and treatment decision tree

Jiaming Gong[1,2☯], Ruimin Zhao[1☯], Zhanhai Yu[3], Jianxue Li[1], Mei Mei[1]*

**1** Department of Stomatology, The 940th Hospital of Joint Logistics Support Force of People's Liberation Army, Lanzhou, Gansu, China, **2** Department of Stomatology, Quzhou Hospital Affiliated to Wenzhou Medical University (Quzhou people's Hospital), Quzhou, China, **3** School/Hospital of Stomatology, Lanzhou University, Gansu, China

☯ These authors contributed equally to this work.
* meim198610@sina.com

**Data Availability Statement:** All relevant data are within the paper and its Supporting information files.

## Abstract

### Background

Implant periapical lesion (IPL), as a peri-implant disease originating from implant apex, maintains coronal osseointegration in the early stage. With the understanding to IPL increasingly deepened, IPL classification based on different elements was proposed although there still lacks an overall classification system. This study, aiming to systematically integrate the available data published in the literature on IPL associated with histopathology, proposed a comprehensive classification framework and treatment decision tree for IPL.

### Methods and findings

English articles on the topic of "implant periapical lesion", "retrograde peri-implantitis" and "apical peri-implantitis" were searched on PubMed, Embase and Web of Science from 1992 to 2021, and citation retrieval was performed for critical articles. Definite histopathology and radiology of IPL are indispensable criteria for including the article in the literature. The protocol was registered in PROSPERO (CRD42022378001). A total of 509 papers identified, 28 studies were included in this review. In only one retrospective study, 37 of 39 IPL were reported to be at the inflammatory or abscess stage. 27 cases (37 implants) were reported, including acute non-suppurative (1/37, developed to chronic granuloma), chronic granuloma (5/37), acute suppurated (2/37), chronic suppurated-fistulized (6/37), implant periapical cyst (21/37), poor bone healing (2/37), foreign body reaction (1/37). Antibiotics alone did not appear to be effective, and the consequence of surgical debridement required cautious interpretation because of the heterogeneity of lesion course and operation. Implant apicoectomy and marsupialization were predictable approaches in some cases.

**Funding:** Our research was supported by the Natural Science Foundation of Gansu Province (20JR10RA006). The funders had no role in study design, data collection and analysis, decision to publish, or preparation of the manuscript.

**Competing interests:** The authors have declared that no competing interests exist.

## Conclusions

The diversiform nature of IPL in the case reports confirms the need for such histopathological classification, which may enhance the comparison and management of different category.

## Introduction

Intraoral implants, possessing the advantage of not affecting the integrity of adjacent teeth and the esthetic properties, have frequently been adopted to complete dentition defects since the concept of osseointegration was put forward [1]. Inevitable implant-related complications have also emerged with implant periapical lesion (IPL) first described as an independent disease entity by McAllister et al in 1992 [2]. Compared to the progression and the affected portion of periimplantitis [3], IPL, originating at the implant tip, maintained normal coronal bone in the early stage.

Previous studies briefly classified IPL as active or inactive in accordance with signs and symptoms [4, 5]. Understandably, active IPL had a tendency to expand and spread proximally, coronally or facially, with localized pain, mucosal swelling, and fistulas. Different etiologies were proposed to play a part in active IPL such as bone overheating during osteotomy, residual infection of the implant bed, and adjacent endodontic lesion [4, 5]. Correspondingly, several terms were used to describe these phenomena, including retrograde periimplantitis, apical periimplantitis and endodontic periimplantitis. Moreover, there also appeared inactive IPL when radiographic manifestations were not relevant to clinical symptoms [4, 5]. The overpreparation of implant bed and the placement of the implant around scar tissue were generally considered to be induced causes.

Of concern was the appearance of a cyst at the implant tip. From the case reports [6–8], implantation may stimulate epithelial residual or inflammatory transformation to formulate cyst at the implant tip, which might cause localized pain, mucosal swelling, and implant mobility. Radiologically, it was indistinguishable from the previously enumerated IPL types. The previously proposed classifications [4, 5, 9–11] (Table 1) omitted implant periapical cyst, which is surprising for the symptom that potentially causes implant failure.

At present, there is a lack of a feasible, comprehensive method to both classify and report all conditions present in implant periapical area. At the end, we proposed a novel classification, in which corresponding treatment decision tree was designed to assess its capability to comprise ever-increasing complexity of manifestation and management. Through a systematic review of the literature evidence, the purpose of this study was to use progressively detailed categories as indicators to describe the multiformity of IPL, explicating latent pathogenesis and treatment protocols.

## Materials and methods

### Protocol

This systematic review complied with the PRISMA statement and its protocol was registered in PROSPERO (CRD42022378001).

### Focus question

The specific research question was: "What histopathological characteristics are associated with IPL?"

**Table 1. The present classification systems regarding the IPL.**

| Author | Category | | Definition |
|---|---|---|---|
| Reiser & Nevins | Inactive (non- infected) | | Apical scar, overdrilling |
| | Active (infected) | | Residual infection or contaminated implant |
| Sussman | Implant to Tooth | | Osteotomy preparation causes adjacent tooth pulp devitalization |
| | Tooth to Implant | | Adjacent tooth periapical pathology or previously existing apical lesion |
| Sarmast et al | Class1, 2 same as Sussman | | Same as Sussman |
| | Improper placement or angulation of the implant | | Implants that are placed too far labially or lingually/palatially |
| | Residual infection | | Residual bacteria/viruses and/or necrotic bone/subclinical infection or placement into an infected or inflamed sinus |
| Penarrocha-Diago et al. | Inactive | Asymptomatic | Apical scar caused by overpreparation or by bone necroses due to overheating |
| | Active | Acute non-suppurated | Acute, spontaneous, continuous pain |
| | | | Mucosa can be swelled and reddish |
| | | | No peri-implant alterations |
| | | Acute suppurated | Implant periapical radiolucency |
| | | | others same as non-suppurated |
| | | subacute or suppurated-fistulized | Dull pain; Periapical radiolucent area |
| | | | Possible fistulous tract or abscess or implant mobility |
| Kadkhodazadeh & Amid | Primary periodontal lesions (P-class) | | P1: apical peri-implantitis |
| | | | P2: marginal peri-implantitis |
| | | | P3: marginal and apical peri-implantitis |
| | Primary Implant complications (I-class) | | I1: apical periodontitis |
| | | | I2: marginal periodontitis |
| | | | I3: marginal and apical periodontitis |
| | Periodontal and peri-implant lesions | | S1: apical lesions |
| | | | S2: marginal lesions |
| | | | S3: marginal and apical lesions |
| | Traumatic lesions with an iatrogenic origin | | T0: non symptomatic |
| | | | T1: symptomatic lesions |
| Shah et al. | Mild | | <25% of the implant length from apex |
| | Moderate | | 25–50% of the implant length from apex |
| | Advanced | | >50% of the implant length from apex |

### Literature searching

The research was performed in accordance with Cochrane Collaboration recommendations, and it included all published articles related to IPL from 1992 to 2021 on PubMed, Embase and Web of Science. Keywords "retrograde peri-implantitis", "apical peri-implantitis", "implant periapical lesion" were searched in the title/abstract, and citation retrieval was performed for critical articles.

### Eligibility criteria

Participants had a history of dental implants. The affected implants needed to be diagnosed as IPL by radiography and histopathology. The results involved radiography, histopathology, and implant outcome. Reviews, conference papers, protocols, non-English publications, and lack of sufficient evidence were excluded.

## Studies selection and data extraction

The studies were independently assessed by two reviewers (Wang and Dai) and disagreements were resolved through discussion. Meanwhile, two reviewers independently extracted the information from the literature according to a preset table, which was then further checked by a third reviewer (ZRM). Domains of extraction included author, number of patients, implant site, follow-up, clinical description, histopathology (extracted verbatim), category, interventions, and outcomes.

## Quality assessment

All case series reports were assessed via modified The Joanna Briggs Institute (JBI) Critical Assessment Checklist (https://synthesismanual.jbi.global.) and the processes were conducted independently by two reviewers (WJ and DZM). The study was assessed as low risk if it provided more than 75% of the required parameters; And parameters of 50% to 75% were assessed as medium risk; Parameter being less than 50% was classified as high risk.

## Information synthesis

The included cases were reviewed for histopathology for the internal integration. Descriptive analysis was used for all extracted information.

# Result

## Searching results

509 literatures were preliminarily searched, and 28 literatures were finally evaluated after eliminating the reduplicative and substandard literatures. (Fig 1).

## Characteristics of the included articles

Table 2 showed the information of IPL extracted after retrieval. There were 27 case reports [6–8, 12–34] and 1 retrospective study [35], all of which were conducted in humans. The following signs and symptoms were frequently mentioned in the case reports: swelling, abscess, localized pain, and fistula. Radiographically, all cases clearly showed radiography in the implant tip, with osseointegration remaining in the implant crown.

## Histopathological assessment

Diverse pathogenesis and progression stages of IPL determine the different histopathological manifestations. In a retrospective study [35], histopathology of 37 implants revealed the intrastromal inflammatory cell infiltration, predominantly lymphocytes and plasma cells, of which 26 implants had indications of infection.

In case reports, infected granulomatous tissue was associated with generalized chronic and acute cells as well as the presence of neutrophils [13, 14, 20]. A great number of inflammatory cells, lymphocyte infiltration and necrotic tissue were observed in abscess [16, 24]. The squamous epithelium may be explored in cystic lesions, partly infiltrated by dense lymphocytes and macrophages, and partly absent from the intraepithelial lining [8, 12, 21, 23]. A case of foreign body was observed by histopathological examination with well-defined starching granules from rubber gloves [22]. Two IPL cases had undesirable bone healing and both, possessing abnormal bone formation and poor trabecular structure, were aseptic [13, 26].

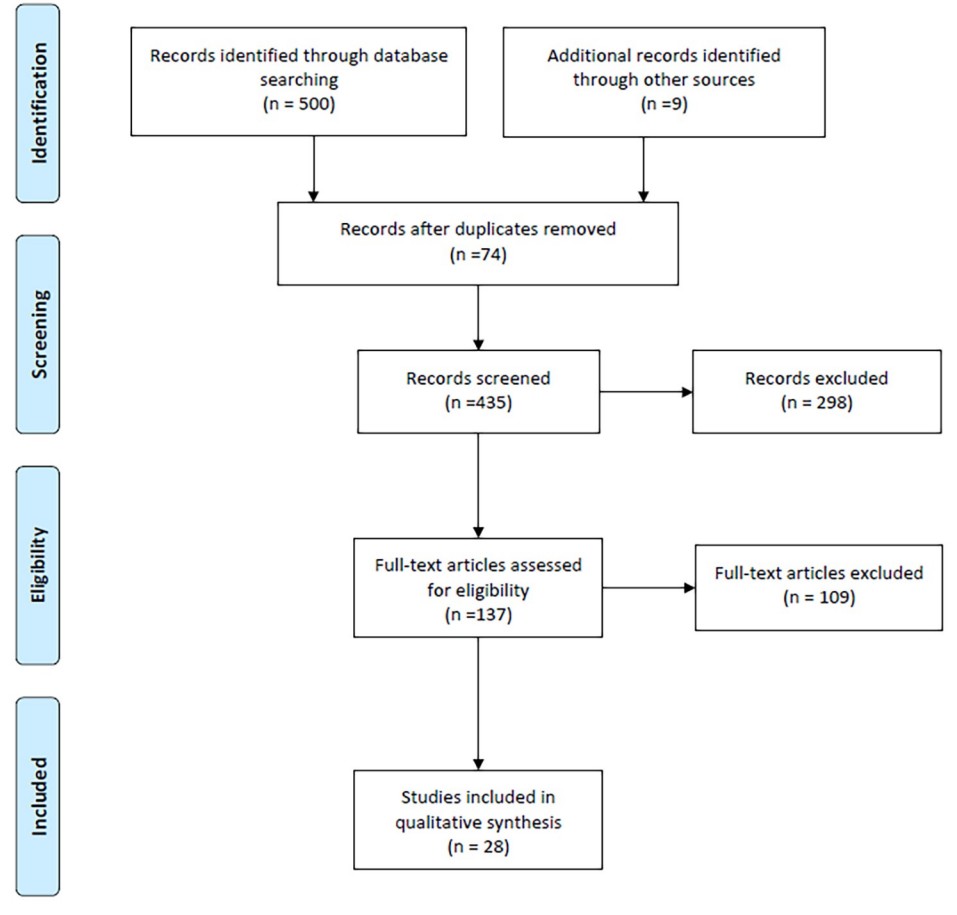

**Fig 1. PRISMA flowchart.**

## Justification for proposed classification

As Table 1 shows, there is still no consensus on the classification of IPL. The included reports (70 patients/76 implants) were preliminarily distributed in Penarrocha-Diago et al. 's classification [5]: 31 implants were inactive [8, 12, 13, 17, 18, 20, 21, 26, 27, 29, 30, 33, 36], and 46 implants were active, with acute non-suppurated(1) [13], acute suppurated (10) [6, 7, 14, 16, 19, 25, 28, 31, 32], subacute/suppurated-fistulized(7) [15, 22–24, 33, 34], unclear stage(26) [35]. It should be noted that histopathological findings in some cases did not match Penarrocha-Diago et al.'s classification to some extent. For example, Nedir et al.'s [22] case presented clinical characteristics similar to the subacute/suppurated-fistulized phase, however, the detected foreign body was considered to be the culprit rather than residual infection; Cases presenting only localized pain and radiography were defined as acute suppurating [6, 7, 31], whereas histopathological evidence showed implant periapical cyst rather than inflammation; Asymptomatic cases were directly classified as inactive lesions according to the previous criteria, which led to the inclusion of cystic entity. However, the interpretation of the inactive item did not comprise cyst.

In view of the above, a more comprehensive and detailed classification was presented in Table 2 and Fig 2. Included case reports were assigned in 5 domains: acute non-suppurated (1/37), chronic granulomatous stage (5/37), acute suppurated (2/37), chronic suppurated-

**Table 2. The characteristics of IPL studies with histological evidence.**

| Author | Patient | Implant site | Follow-up | Clinical description | | | | | Radiolucency | Histopathology (verbatim extracts) | Category | Interventions | | Outcomes |
|---|---|---|---|---|---|---|---|---|---|---|---|---|---|---|
| | | | | Swelling | Pain | Abscess | Fistula | Mobility | | | | Non-surgery | Surgery | |
| Balshi et al. 2007 | Case 39 | Maxilla (9 anterior, 8 posterior); mandible (11 anterior, 11 posterior) | Average of 1.64 years | Only 66.7% (26 of 39) demonstrated clinical evidence of infection (eg, swelling, suppuration, fistula formation). | | | | - | 39+ | ...a stroma of delicate bundles of immature collagen fibers interspersed by active fibrocytes and numerous dilated capillaries. Throughout the stroma an infiltrate of inflammatory cells, predominantly lymphocytes and plasma cells, was reported. | Not sure | - | Apicoectomy and GBR | One failure; 38 success |
| Casado et al. 2008 | Case one | 11 | 3 years | - | - | - | - | - | + | In the cyst wall a dense lymphocytic infiltrate, veins, and peripheral nerves... | Implant periapical cyst | - | Organic bovine matrix graft | Success |
| Chan et al. 2011 | Case two | 41 | 3 months | - | - | - | - | - | + | ...fibrous connective tissue with a mild chronic mixed inflammatory cell infiltrate...clusters of inflammatory cells, primarily lymphocytes, were noted in a background of relatively dense connective tissue. | Poor bone healing | - | Bone graft (allograft) | Success |
| | | 24 | 1 month | + | - | - | - | - | + | ...numerous acute and chronic inflammatory cells could be seen in a background of immature granulation tissue containing numerous small to medium-diameter blood vessels... | Chronic Granuloma | RCT of adjacent tooth | GBR with cortical bone allograft and bioabsorbable membrane | Success |
| Chaffee et al. 2001 | Case one | 46 | 35 days | - | - | + | - | - | + | ...the presence of granulation tissue containing acute inflammatory cells, necrotic debris, and abundant hemorrhage bound by fibrous connective tissue infiltrated with chronic inflammatory cells. | Chronic suppurated-fistulized | RCT of adjacent tooth | GBR with DFDBA and barrier | Success |

(*Continued*)

**Table 2.** (Continued)

| Author | Patient | Implant site | Follow-up | Clinical description | | | | | | Histopathology (verbatim extracts) | Category | Interventions | | Outcomes |
|---|---|---|---|---|---|---|---|---|---|---|---|---|---|---|
| | | | | Swelling | Pain | Abscess | Fistula | Mobility | Radiolucency | | | Non-surgery | Surgery | |
| Dahlin et al. 2008 | Case one | 43 | 2 years | + | + | - | + | - | + | Granulation tissue… revealed a periapical inflammatory infection around the top of the implant. | Chronic suppurated-fistulized | - | Apicoectomy | Success |
| Favia et al. 2011 | Case one | 36 | 4 months | - | - | + | - | - | + | …a gap was observed between bone and implant. This bone was nonvital, and many osteocyte lacunae were empty…. No newly formed bone or osteoblasts…No osteoclasts or Howship lacunae… to see a connective tissue with an inflammatory cell infiltrate | Chronic suppurated-fistulized | - | Debridgement | Failure |
| | | 37 | 4 months | - | - | + | - | - | + | …bone trabeculae were observed within the apical implant threads… Osteoid matrix was present in many portions; no osteoblasts…a loose connective tissue with many spindle cells, plasma cells, and many inflammatory cells… | Chronic suppurated-fistulized | - | Debridgement | Failure |
| Galzignato et al. 2010 | Case one | 12 | 3 months | - | - | - | - | - | + | …a residual odontogenic inflammatory cyst, characterised by a thick, irregular, often incomplete, squamous epithelium, with granulation tissue forming the cyst wall in the denuded areas. The fibrous capsule… | Implant periapical cyst | - | Remove cyst | Failure |

(Continued)

**Table 2.** (Continued)

| Author | Patient | Implant site | Follow-up | Clinical description | | | | | | Histopathology (verbatim extracts) | Category | Interventions | | Outcomes |
| | | | | Swelling | Pain | Abscess | Fistula | Mobility | Radiolucency | | | Non-surgery | Surgery | |
|---|---|---|---|---|---|---|---|---|---|---|---|---|---|---|
| Kochaji et al. 2017 | case one | 46 | 6 months | - | - | - | - | - | + | . . . a layer of inconspicuous nonkeratinized stratified squamous epithelium lying on an inflamed fibrous tissue wall with a dense capsule-like outerlayer. The epithelial nature of the lining cells was confirmed. . . a radicular or apical inflammatory dental cyst around the apex of a tooth. | Implant periapical cyst | - | Remove cyst | Failure |
| | case one | 46 | 9 months | - | - | - | - | - | + | . . . a cyst wall with the lumen lined by hyperplastic non-keratinized epithelium of several cell layers thickness supported by immature and mature fibrous tissue. . . a radicular cyst. | Implant periapical cyst | - | Remove cyst | Failure |
| Kim et al. 2013 | Case one | 15,16,17 | 10 years | + | - | - | - | - | + | . . . a pseudostratified ciliated columnar epithelium and a partly stratified squamous epithelium. There were some inflammatory cells in the cyst wall. A pathologic diagnosis of a POMC was made. | Implant periapical cyst | - | Remove cyst and bone graft (xenograft and autogenous cortical bone) | Success |
| Manfro et al. 2018 | Case one | 21 | 1 year | - | + | - | - | - | + | . . . The implant was not osseointegrated in the chronically infected apical alveolar bone. | Chronic Granuloma | Antibiotics -cephalexin | Debridgement (EDTA, Ethylene Diamine Tetraacetic Acid), apicoectomy and bone graft (alloplastic biphasic calcium phosphate material) | Success |

(*Continued*)

Table 2. (Continued)

| Author | Patient | Implant site | Follow-up | Clinical description | | | | | Radiolucency | Histopathology (verbatim extracts) | Category | Interventions | | Outcomes |
|---|---|---|---|---|---|---|---|---|---|---|---|---|---|---|
| | | | | Swelling | Pain | Abscess | Fistula | Mobility | | | | Non-surgery | Surgery | |
| Mccracken et al. 2012 | Case one | 42 | 2 years | - | - | - | - | - | + | The biopsy was read as a periapical granulma with generalized chronic and acute inflammation, with associated vital reactive bony spicules. | Chronic Granuloma | - | Debridgement | Success |
| Mccrea et al. 2014 | Case one | 21 | 3 years | - | - | - | - | - | + | The fibrous wall was lined by thin, stratified squamous epithelium and partly by pseudostratified columnar epithelium and cuboidal epithelium. A few nerve bundles and blood vessels were also present in the wall. | Implant periapical cyst | - | Remove cyst and GBR with allograft, Bio-Oss and Bio-gide | Success |
| Nedir et al. 2007 | Case one | 15 | 3.5 years | + | + | + | + | - | + | The starch distribution in the tissue was not homogeneous; starch particles seemed to agglomerate. | Foreign body reaction | Amoxicillin presurgury | Apicoectomy | Sucesss |
| Piattelli et al. 1995 | Case one | 14 | 2 months | + | + | + | - | - | + | . . .small colonies of bacteria around the outer perimeter of the implant. . .tissue that stained with basic fuchsin inside the hole in the apical part of the implant | Acute suppurated | Antibiotic partially resolution | FDDMA membrane | Failure |
| Piattelli et al. 1998 (1) | Case one | 14 | 7 months | - | - | - | - | - | + | . . . necrotic bone was observed inside the antirotational hole; all of the osteocyte lacunae were empty. . . . The bone trabeculae appeared to be compressed, and some of them had undergone demineralization. . . All other parts of the implant surface were surrounded by vital, compact, mature bone. | Poor bone healing | Metronidazole | Debridgement | Failure |

(Continued)

**Table 2.** (Continued)

| Author | Patient | Implant site | Follow-up | Clinical description | | | | | | Histopathology (verbatim extracts) | Category | Interventions | | Outcomes |
|---|---|---|---|---|---|---|---|---|---|---|---|---|---|---|
| | | | | Swelling | Pain | Abscess | Fistula | Mobility | Radiolucency | | | Non-surgery | Surgery | |
| Piattelli et al. 1998 (2) | Case one | Premolar of right mandible | 5 months | + | + | - | + | - | + | …bone and intiammotory tissue with an absence ot vasculor structures. The inflammotory cell infiltrôte…showed a prevaience ot mocrophages and iymphocytes, with piasma cells and granulocytes … | Chronic suppurated-fistulized | Metronidazole | Debridgement | Failure |
| Park et al. 2004 | Case one | 15,17 | 13 years | + | - | - | - | - | + | … as POMC. Some cilia were observed, but ciliary loss due to chronic inflammation was also evident. The cystic lesion was lined with pseudostratified columnar epithelial cells… | Implant periapical cyst | - | Remove cyst | One failure; one success |
| Pistilli et al. 2020 | Case three | 12 | 4 years | + | + | - | - | - | + | The histopathologic report described the lesion as a cyst measuring 7*5 millimeters. | Implant periapical cyst | - | Remove cyst, apicoectomy and GBR | Success |
| | | 36 | 1 year | - | + | - | - | - | + | The histopathologic report described the lesion as a cyst measuring 8*5 mm. | Implant periapical cyst | - | Remove cyst and GBR | Success |
| | | 47 | Not report | + | + | - | - | - | + | The histopathologic report described the lesion as a cyst measuring 15*9 mm. | Implant periapical cyst | - | Remove cyst and GBR | Success |
| Qu et al. 2013 | Case one | 46 | 5 months | - | - | - | - | - | + | …inflammatory cyst wall-like lesion, with the infiltration of macrophage and lymphocytes, but the epithelial lining was undetected. | Implant periapical cyst | - | Cyst removal | Success |

(*Continued*)

**Table 2.** (Continued)

| Author | Patient | Implant site | Follow-up | Clinical description | | | | | | Histopathology (verbatim extracts) | Category | Interventions | | Outcomes |
|---|---|---|---|---|---|---|---|---|---|---|---|---|---|---|
| | | | | Swelling | Pain | Abscess | Fistula | Mobility | Radiolucency | | | Non-surgery | Surgery | |
| Scarano et al. 2000 | Case one | Premolar in right mandible | 6 months | - | + | - | - | - | + | …bone and non mineralized tissues were present only in the most apical portion of the implant….…necrotic and almost completely demineralized bone was present; some multinucleated cells … | Chronic Granuloma | - | Debridgement | Failure |
| Silva et al. 2010 | Case one | 21 | 1 year | + | - | - | - | - | + | The lesion and implant were completely removed, and histological examination confirmed the diagnosis of periapical inflammatory cyst. | Implant periapical cyst | - | Remove cyst | Failure |
| Sun et al. 2013 | Case one | 35 | 2.5 years | - | + | + | - | - | + | …2 sulfur granules demonstrating granular and fibrillar basophilic to amphophilic bacterial colonies associated with peripheral purulent exudates (neutrophils)… | Acute suppurated | - | Debridgement with tetracycline | Failure |
| Sivolella et al. 2013 | Case one | 11,12 | 5 years | - | - | - | - | - | + | …the cyst was covered with a layer of epithelium comprising three epithelial cell types, i.e., ciliated columnar (respiratory), cuboidal, and non-keratinised stratified squamous epithelium. … | Implant periapical cyst | - | Remove cyst and GBR | One failure; one success |
| Sukegawa et al. 2014 | Case one | 11 | 9 years | + | + | - | - | - | + | … the cyst wall was lined with either stratified squamous epithelium or columnar epithelium. The cyst wall consisted of fibrous connective tissue, and relatively large vessels and nerves were observed… | Implant periapical cyst | - | Remove cyst | Success |

(*Continued*)

**Table 2.** (Continued)

| Author | Patient | Implant site | Follow-up | Clinical description | | | | | Radiolucency | Histopathology (verbatim extracts) | Category | Interventions | | Outcomes |
|---|---|---|---|---|---|---|---|---|---|---|---|---|---|---|
| | | | | Swelling | Pain | Abscess | Fistula | Mobility | | | | Non-surgery | Surgery | |
| Takesshita et al. 2013 | Case one | 21 | 2.5 years | - | + | - | - | - | + | … the wall of the cystic lesion comprised of cuboidal, ciliated columnar and stratified squamous epithelium with underlying connective tissue… | Implant periapical cyst | - | Remove cyst and apicoectomy | Success |
| Tseng et al. 2005 | Case one | 45 | 6 months | - | - | - | - | - | + | The curetted apical tissue was sent for pathology diagnosis, and a radicular cyst was subsequently diagnosed. | Implant periapical cyst | - | Remove cyst | Failure |
| Thompson-Sloan et al. 2012 | Case one | 11 | 10 years | - | - | - | - | - | + | …revealed predominantly fibrovascular | Chronic Granuloma | Antibiotics-clindamycin | Apicoectomy and GBR with demineralized bone matrix and collagen barrier | Success |
| | | 21 | 10 years | - | - | - | + | - | + | connective tissue and granulation tissue… | Chronic suppurated-fistulized | | | |
| Troiano et al. 2020 | Case one | 44 | No report | + | - | - | + | - | + | …the cystic wall was covered by a stratified non-keratinizing squamous epithelium …. the epithelium appeared hyperplastic, with acanthosis, vacuolization of the cheratinocytes, and focal granulocyte exocytosis. | Implant periapical cyst | RCT of adjacent tooth | Marsupialization | Success |

+, present; -, absent; NRRCT, Root canal therapy; GBR, Guided bone regeneration

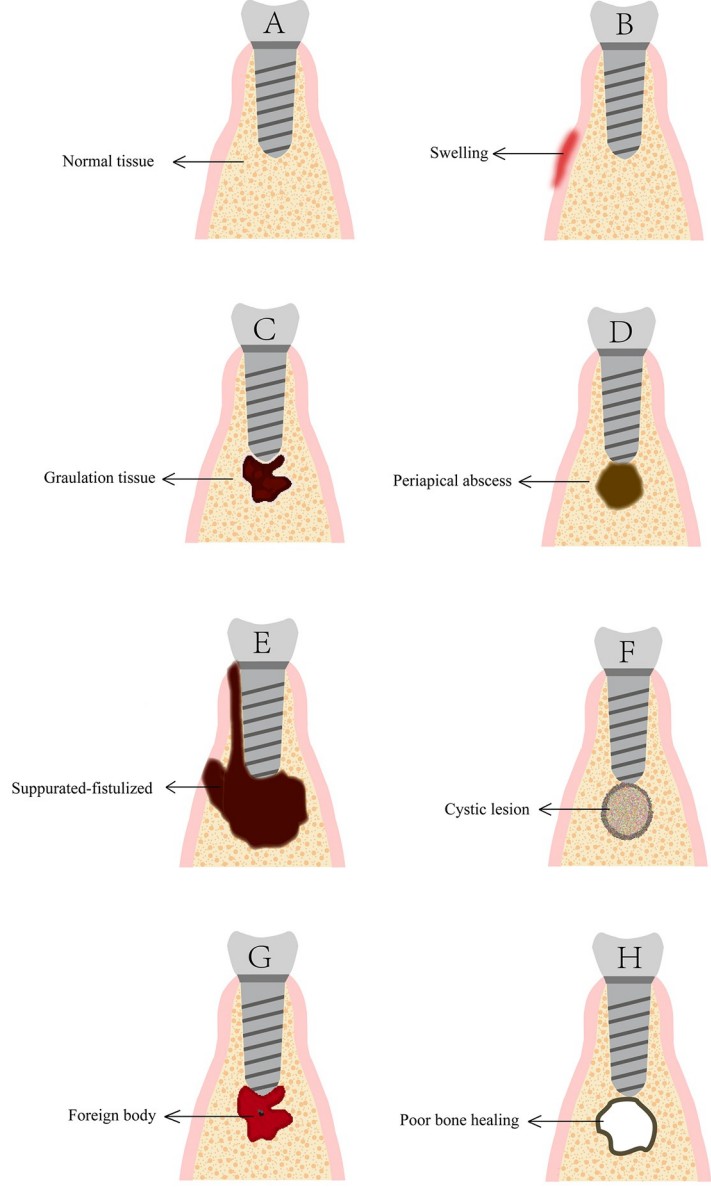

**Fig 2. Schematic representation of each category of histological IPL classification.** A: Normal implant periapical status; B: Acute non-suppurative stage (no radiological changes); C: Chronic granulomatous stage; D: Acute suppurative stage (pus formation); E: Chronic suppurative fistula (two-drainage pathways); F: Implant periapical cyst. G: Foreign body reaction (radiologically visible or invisible); H: Poor bone healing.

fistulized (6/37), implant periapical cyst (21/37), foreign body reaction (1/37), poor bone healing (2/37). The retrospective study was not involved in the above statistics due to the lack of individual case content.

## Class I. Implant periapical inflammation

1. Acute non-suppurated (Fig 2B) [5]:

   - Symptoms or signs: Acute, continuous, moderate to severe and localized pain; Not aggravating the pain with percussion; Perhaps painful and inflamed periapical mucosa;

- Histopathology: Acute inflammatory infiltrate, neutrophil infiltration

- Radiography: No radiolucency

2. Chronic granuloma (Fig 2C):

- Symptoms or signs: No symptoms or light spontaneous pain; Perhaps swelled and reddish mucosa

- Histopathology: Inflammatory granulation tissue; Increased inflammatory cells and capillaries

- Radiography: Radiolucency

Class II. Implant periapical abscess

1. Acute suppurated (Fig 2D) [5]:

- Symptoms or signs: Same as the non-suppurated case

- Histopathology: Polymorphonuclear leukocytes infiltrate and necrotic tissue

- Radiography: Radiolucency without penetrating the bone-plate

2. Chronic suppurated-fistulized (Fig 2E) [5]:

- Symptoms or signs: Dull pain; Possible sinus tract around the Mucosa; A tympanic sound produced by percussion

- Histopathology: Fibrous connective tissue hyperplasia or infiltration of lymphocytes

- Radiography: Radiolucency with possible incomplete bone-plate

   Class III. Implant periapical cyst (Fig 2F):

- Symptoms or signs: No symptoms and possible mucosa swelling

- Histopathology: Epithelial lining and possible cholesterol crystallization

- Radiography: Radiolucency

   Class IV. Foreign body reaction (Fig 2G):

- Symptoms or signs: Inflammatory response or no symptoms

- Histopathology: Foreign bodies

- Radiography: Radiolucency or not

   Class V. Poor bone healing (Fig 2H):

- Symptoms or signs: No symptoms

- Histopathology: Aseptic necrosis or fibrous connective tissue

- Radiograph: Radiolucency

## Treatment protocols and results

Variability of the properties did not allow the management of individual patients to be discerned, so the treatment protocols in case reports were rearranged according to the new classification format.

1. Acute non-suppurated (1 case): by the time of intervention, the acute non-suppurative stage had progressed to the chronic granulomatous stage [13].

2. Acute suppurated (2 cases): One case, failing to respond to antibiotics, was covered with collagen membrane after surgical debridement [25]. Tetracycline was disinfected after direct surgical debridement in another case [32]. Both implants ultimately failed to survive.

3. Chronic granuloma (5 cases): Two cases did not respond to antibiotics and were treated with implant apicoectomy and bone graft materials [19, 33]. One case performed RCT on the adjacent teeth with the same surgical protocol as the above [13]. In the other two cases, one failed while the other one survived after implant debridement [20, 28].

4. Chronic suppurated-fistulized (6 cases): Antibiotic treatment failed in both cases [24, 33], followed by the implant failure after surgical debridement in one case and the success with apicoectomy plus GBR in the other. One case was successfully treated with surgical debridement plus GBR and adjacent RCT [14]. The implant was not survived after surgical debridement in two cases [16], but one was reserved by implant apicoectomy [15].

5. Implant periapical cyst (21 cases): Only the cysts were removed by surgical debridement of the 9 implants [8, 17, 23, 27, 29, 31, 36], resulting in 6 falling and 3 remaining. Apicoectomy was performed in two implants [6, 23] and marsupialization was performed in one [34]. At the end, all implants survived.

6. Poor bone healing (2 cases): After debridement of the two implants, one implant healed well [13] while the other implant was removed because of the existing lesion [26].

In the retrospective study [35], 39 IPL were undergoing with apicoectomy. Ultimately, only 1 implant failed.

## Possibility of bias

Nine papers were assessed as low possibility of bias, 13 as moderate possibility of bias, and 6 as high possibility of bias (Table 3).

## Discussion

Replacing the space among the missing teeth with implants is the best alternative to restore the patient's oral morphology and function. However, complex and varied peri-implant diseases generally affect the long-term outcomes of the implant [37]. Early studies mentioned the loss of periapical supporting bone in implants, which was presumed to be related to microbial residue, bone overheating and premature loading [38, 39]. At present, IPL is considered to possess the multifactorial induction, with adjacent endodontic lesion having the highest priority [40]. Given the complexity and uncertainty of the pathogenesis, the incomplete recognition of IPL has led to the limitations of previous classification systems.

Reiser and Nevins [4] primarily divided IPL into inactive and active forms, which were also employed by Sarmast et al. [9] and Penarrocha-Diago et al [5]. Differently, the former increased the category of implant misplacement and residual infection from etiological consideration. The latter refined the characteristics of inflammatory stage from the stage of lesion progression. Regrettably, the above classifications failed to assess the histopathology of IPL, resulting in the neglection of cyst entities. Sussman et al. [41] believed that the categories "Implant to Tooth" and "Tooth to Implant" explained the potential mutual relationship between adjacent teeth and IPL. This inference was based on radiography, but it was undeniable that IPL sometimes occurred independently from the adjacent teeth. Besides,

**Table 3. JBI critical appraisal for articles.**

| JBI for case reports | Were patient's demographic characteristic clearly described? | Was the patient's history clearly described and presented as a timeline? | Was the current clinical condition of the patient as a presentation clearly described? | Were the diagnostic tests or assessment methods and the results clearly described? | Was the intervention(s) or treatment procedure(s) clearly described? | Was the post-intervention clinical condition clearly described? | Were the adversed events(harms) or unanticipated events identified and described? | Does the case report provide takeaway lessons? |
|---|---|---|---|---|---|---|---|---|
| Balshi et al. 2007 | Y | N | N | N | Y | Y | N | Y |
| Casado et al. 2008 | Y | N | N | Y | Y | N | Y | Y |
| Chan et al. 2011 | Y | N | Y | N | Y | Y | N | Y |
| Chaffee et al. 2001 | N | N | N | Y | Y | Y | Y | Y |
| Dahlin et al. 2008 | Y | Y | Y | Y | Y | Y | N | Y |
| Favia et al. 2011 | Y | N | N | Y | Y | Y | N | N |
| Galzignato et al. 2010 | Y | Y | N | Y | N | N | N | Y |
| Kochaji et al. 2017 | Y | Y | N | Y | N | N | N | N |
| Kim et al. 2013 | Y | Y | N | Y | N | Y | N | Y |
| Manfro et al. 2018 | N | N | N | Y | Y | Y | Y | Y |
| Mccracken et al. 2012 | N | N | N | Y | Y | Y | N | Y |
| Mccrea et al. 2014 | N | N | Y | Y | Y | Y | N | Y |
| Nedir et al. 2007 | Y | N | Y | Y | Y | Y | Y | Y |
| Piattelli et al. 1995 | N | N | N | Y | Y | N | N | Y |
| Piattelli et al. 1998 (1) | N | N | N | N | N | N | Y | Y |
| Piattelli et al. 1998 (2) | N | N | N | N | N | N | Y | N |
| Park et al. 2004 | Y | Y | N | Y | Y | Y | N | Y |
| Pistilli et al. 2020 | Y | Y | N | Y | Y | Y | N | Y |
| Qu et al. 2013 | N | N | Y | Y | Y | Y | Y | Y |
| Scarano et al. 2000 | N | N | N | Y | N | N | Y | Y |
| Silva et al. 2010 | Y | N | N | Y | N | Y | N | Y |

(*Continued*)

**Table 3.** (Continued)

| JBI for case reports | Were patient's demographic characteristic clearly described? | Was the patient's history clearly described and presented as a timeline? | Was the current clinical condition of the patient as a presentation clearly described? | Were the diagnostic tests or assessment methods and the results clearly described? | Was the intervention(s) or treatment procedure(s) clearly described? | Was the post-intervention clinical condition clearly described? | Were the adversed events(harms) or unanticipated events identified and described? | Does the case report provide takeaway lessons? |
|---|---|---|---|---|---|---|---|---|
| Sun et al. 2013 | Y | Y | Y | Y | Y | Y | Y | Y |
| Sivolella et al. 2013 | Y | Y | N | Y | N | Y | N | Y |
| Sukegawa et al. 2014 | Y | Y | N | Y | Y | N | N | Y |
| Takesshita et al. 2013 | Y | N | Y | Y | Y | Y | N | Y |
| Tseng et al. 2005 | N | N | N | Y | Y | N | N | Y |
| Thompson-Sloan et al. 2012 | N | N | N | Y | Y | Y | Y | Y |
| Troiano et al. 2020 | Y | Y | N | Y | Y | Y | N | Y |
| JBI for cohort study | Were there clear criteria for inclusion in the case series? | Was the condition measured in a standard, reliable way for all participants included in the case series? | Were valid methods used for identification of the condition for all participants included in the case series? | Did the case series have complete inclusion of participants? | Was there clear reporting of clinical information of the participants? | Were the outcomes or follow up results of cases clearly reported? | Was there clear reporting of the presenting site(s)/clinic(s) demographic information? | Was statistical analysis appropriate? |
| Balshi et al. 2007 | Y | Y | Y | N | N | Y | Y | Y |

Kadkhodazadeh and Amid [11] proposed a complex classification of peri-implant disease focusing on the relationship between adjacent teeth and implant, but the interpretation of IPL was limited. Recently, Shah et al. [10] proposed a quantitative classification employing radiology to measure the affected proportion of implant. The failure to consider the pathogenesis of IPL made this classification in the need of combing with other classifications.

In this context, the proposed classification provided a comprehensive description of IPL and offered the potential to increase our knowledge and understanding of management. In Table 2, the present possible entity of IPL rather than ordinary inflammatory property was indicated, especially the histopathological evidence of cyst.

For this classification, each category has been subjected to rigorous literature screening and scholars' evaluation while referring to Penarrocha-Diago's proposals [5]. Within the category "implant periapical inflammation", acute non-suppurative phases were less common in the included studies. The localized pain around the implant tip aroused the attention of the implantologists, but the apical radiography could not be observed. Chan et al. [13] reported that the acute suppurative stage progressed to the chronic granuloma stage, which could be detected by apical radiograph a few days after prophylactic antibiotic administration. Apart from radiological differences, the pain response of the former was usually more severe than that of the latter, which showed granulation tissue on biopsy.

The contents of implant periapical abscess described in Penarrocha-Diago's review [5] were confirmed by the histopathology of the included studies. The limited number of cases was due to the exclusion of previous cases focusing on radiology and ignoring histopathology. As a matter of fact, cases at this stage were often described. Pain is particularly intense during this acute suppurative stage. Without early intervention, infection can spread along the implant-bone interface, ultimately leading to the implant failure. It can also spread the facial bone-plate and form mucosal fistulas penetrated with oral cavity, allowing oral microbial infiltration. The similarity of Penarrocha-Diago's results provides a degree of assurance with which possible histopathology was integrated.

The term "implant periapical cyst" was first mentioned and reviewed in this classification. Multiple studies reported that implant placement induced cyst formation in implant tip [6, 8, 30, 31]. As described in the case reports, the stimulation of infection or implantation might stimulate epithelial proliferation to form the cyst wall, which was composed of squamous epithelium and could be observed in tissue sections with or without inflammatory cell infiltration, depending on the origin of cyst. In general, patients with implant periapical cysts feel normal in the early stage, but the compression of enlarged cyst can cause various symptoms, such as mucosal swelling, local pain, fistula, and implant mobility. Although cysts (21/35) are the most frequently documented in Table 2, the prevalence is underestimated because some professionals may surgically remove implant periapical tissue without histopathological examination. As it can be difficult to clinically differentiate between implant periapical cyst and other IPL, the definitive diagnosis based on the histopathology study is significant.

Foreign body reaction refers to the inflammatory response caused by the presence of foreign materials at the implant tip that affects bone healing [42]. Foreign bodies, such as glove powder and metal particles from instruments, are usually brought in during the implantation. Nedir et al. [22] examined starch particles in rubber gloves in IPL that caused localized chronic granuloma or delayed hypersensitivity. Radiologically, foreign body reaction did not always present radiolucency, posing a conundrum to distinguish it from initial inflammation.

The incorporation of scar tissue, overpreparation of implant bed, and bone compression for inclusion [40] into the proposed term "poor bone healing" were considered. They are often clinically asymptomatic and radiologically confused with implant periapical cyst, therefore, aseptic, non-cystic histopathological diagnosis is critical. Hence, poor bone healing is not a

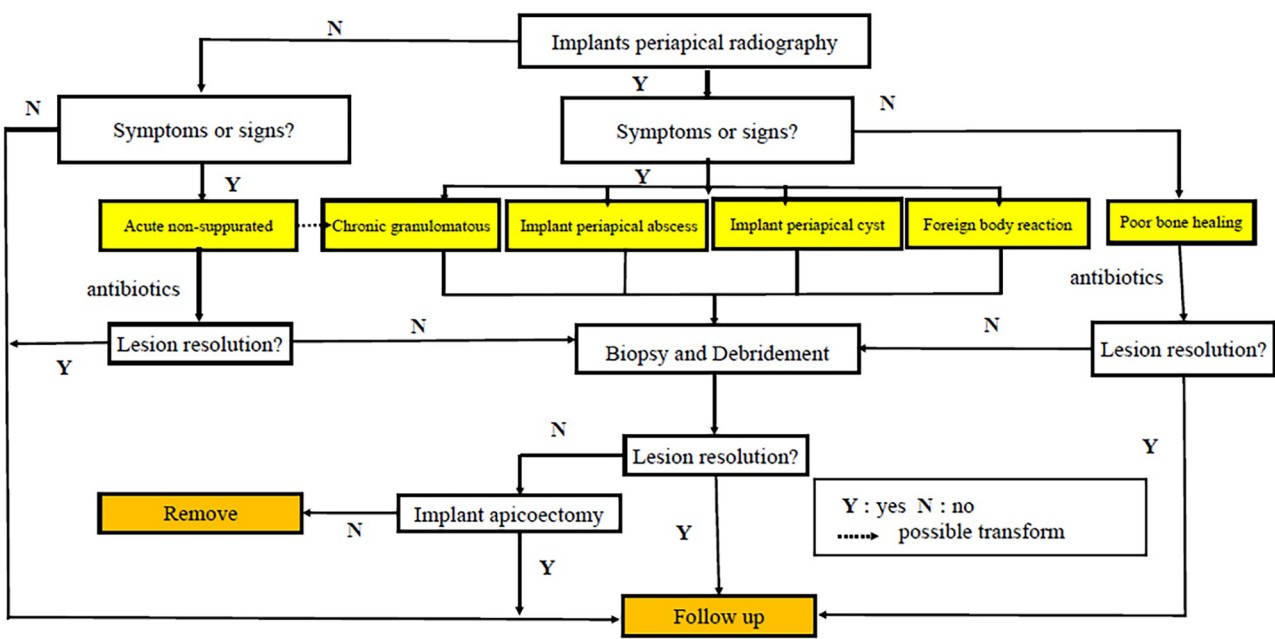

**Fig 3. Treatment decision tree.**

disease or pathological condition in the new classification. The term is simply defined as "inactive" based on the clinically asymptomatic features and stable lesion ranges seem to require greater caution.

Cases in the literature reported that the treatment of IPL, as an empirical approach rather than the types of IPL (single or multiple), was treated by a particular technique. Treatment decision tree (Fig 3) was attempted to be reported by organizing the same category of information in the IPL. Poor bone healing with radio-only transmission was shown in Fig 2H, which has been suggested by several studies to monitor lesions without medical intervention [43]. Once the radiography increases or the patient experiences pain, surgery is required [43]. In the acute non-suppurative phase shown in Fig 2B, experimental antibiotics seem to be a reasonable conservative option to observe the progression of disease [44], although enrolled cases suggest a great possibility of failure. Reviewing the unincluded literature, it was found that systemic antibiotics for IPL included clindamycin, metronidazole, amoxicillin, cephalexin, penicillin, and cephalosporin [25, 26, 45–48]. Among them, Waasdorp et al [49] and Chang et al [50] respectively used amoxicillin (500mg/d, 10d), amoxicillin (250mg/d, 3d) and acetaminophen(500mg/d, 3d), achieving a surprising success without additional management. At present, there is no consensus on the dose and type of antibiotics for IPL, and its criteria should consider etiology, symptom and open / closed lesions [50].

Symptoms (localized pain and puffiness) and signs (mucosal swelling and fistulas) may occur at various intensifications in different stages of infection. Foreign body reaction is shown in Fig 2G. Implant periapical radiography further confirms the need for surgical intervention [44]. Thorough debridement is identified as the centroid for the prognosis of implant, especially the plaque biofilm on the rough surface of implant [43]. Implant apicoectomy is considered to be prudent because the elevated crown/implant ratio increases the risk of unexpected mechanical complications, although it is currently considered as the most thorough and successful procedure in clinical practice (44/45). Conservative implant surface preparation

has been reported, including mechanical curettage [13, 25], chemical agents [32], air-abrasive and laser decontamination [51], whereas limited case results suggest that there is no standardized prospective protocol. In this context, a phased debridement protocol was recognized, in which conservative non-resectable surgery was given priority and apicectomy was considered after ineffectiveness [40, 43]. Besides, the healing of soft tissue involved in IPL are also of concern, especially in cases of mucosa fistula.

To our knowledge, this is the first comprehensive consideration of implant periapical cyst that reduces the diagnostic complexity of different types of cysts and improves the chances of clinical use. The protocols for removing cyst are not exactly the same as the infection, with the primary privilege concentrating on the treatment of giant implant periapical cyst. While surgical excision was only discussed in the previous study, Troiano et al. [34] provided a potential solution for the successful treatment of a large implant periapical cyst with marsupialization. Biopsy during surgery is considered as a necessary element to identify recurrent cystic or malignant tumors [8, 17]. Obviously, this effectively reduces the risk of complications such as implant mobility and fracture caused by direct debridement, thus resulting in large scale of bone defects.

In conclusion, this report, presenting a general classification framework that can highlight the complexity of IPL, is suitable for integrating into the clinical practice. We have done preliminary verification with limited evidence. However, additional cohort studies containing histopathological evidence are necessary to complement and refine the applicability and comprehensiveness of the new classification. This classification is timely for IPL although the ointment is the inability to verify the optimal treatment protocols.

## Supporting information

**S1 File.**
(ZIP)

## Acknowledgments

Authors are thankful for Dr. Wang and Dr. Dai to help with the literature screening and quality assessment.

## Author Contributions

**Conceptualization:** Jiaming Gong.

**Data curation:** Ruimin Zhao.

**Formal analysis:** Ruimin Zhao.

**Funding acquisition:** Jianxue Li.

**Methodology:** Ruimin Zhao.

**Project administration:** Jianxue Li.

**Supervision:** Zhanhai Yu.

**Validation:** Zhanhai Yu.

**Visualization:** Zhanhai Yu.

**Writing – original draft:** Jiaming Gong.

**Writing – review & editing:** Mei Mei.

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
