## [Decision Letter · Decision Letter 0]

21 Nov 2022

PONE-D-22-29419A Novel Histopathological Classification of Implant Periapical Lesion: A Systematic Review and Treatment Decision TreePLOS ONE

Dear Dr. Mei Mei

Thank you for submitting your manuscript to PLOS ONE. After careful consideration, we feel that it has merit but does not fully meet PLOS ONE’s publication criteria as it currently stands. Therefore, we invite you to submit a revised version of the manuscript that addresses the points raised during the review process.

Please review  the comments carefully and do appropriate corrections as advised by the  peers.

We look forward to receiving your revised manuscript.

Kind regards,

Fahad Umer

Academic Editor

PLOS ONE

2. Please ensure that you include the complete search strategy in your manuscript, or as a Supplementary Information file, such that it could be repeated.

“This study was supported by the following funding:

The Natural Science Foundation of Gansu Province (20JR10RA006)”

7. Please include a separate caption for each figure in your manuscript.

8. Please include your tables as part of your main manuscript and remove the individual files. Please note that supplementary tables (should remain/ be uploaded) as separate "supporting information" files.

Reviewers' comments:

Reviewer's Responses to Questions

**Comments to the Author**

1. Is the manuscript technically sound, and do the data support the conclusions?

Reviewer #1: Yes

Reviewer #2: Yes

2. Has the statistical analysis been performed appropriately and rigorously? 

Reviewer #1: Yes

Reviewer #2: N/A

3. Have the authors made all data underlying the findings in their manuscript fully available?

Reviewer #1: Yes

Reviewer #2: Yes

4. Is the manuscript presented in an intelligible fashion and written in standard English?

Reviewer #1: Yes

Reviewer #2: Yes

5. Review Comments to the Author

Reviewer #1: The concept of this systematic review is clinically applicable and will add to the literature to improve our understanding of the implant periapical lesions.

There are few areas that need clarification.

1. Why was this review not registered at Prospero ?

2. Proper analysis method was not adapted using PICO.

3. Since antibiotics didn't play a strong role in resolution of the lesion, then why was it still advocated in the treatment decision tree ?

4. In case of the antibiotics usage, what was the dosage and type recommended ?

5. Were all the included patients healthy ?

Reviewer #2: Overall its an interesting systematic review.

In proposed classification and management, the Author did not take into account soft tissue changes. for example fig 2e could propose soft tissue challenges.

It is a systematic review therefore statistical analysis is not applicable.

Minor typing and grammatical errors.

In result, you did not report the quality you assign to each study. Either summarised in the results section or as an extra column in your study characteristics table.

Most of the studies have one or two cases therefore any management recommendation should be advised cautiously.

Any recommendation on antibiotics protocols, such as before or after procedure, and what antibiotics to be considered?

6. PLOS authors have the option to publish the peer review history of their article (what does this mean?). If published, this will include your full peer review and any attached files.

Reviewer #1: No

Reviewer #2: No

---

## [Author Response · Author response to Decision Letter 0]

5 Dec 2022

Dear editor and reviewers:

We sincerely thank the editors and reviewers for their careful review and detailed suggestions on this manuscript, which undoubtedly enriches the article. The following is our reply to the revision of the article, and the revised parts are underlined in the manuscript:

Response to editor Fahad Umer:

1. Please ensure that your manuscript meets PLOS ONE's style requirements

Answer: We apologize for the nonstandard format of our manuscript. Through the link you provided, we have made modifications according to the requirements, including abstract, font size, references, title, etc. We hope this modification can meet the requirements of the journal. Please contact us in time if anything is still not suitable.

2. Please ensure that you include the complete search strategy in your manuscript, or as a Supplementary Information file, such that it could be repeated.

Answer: We sincerely acknowledge that this was an oversight on our part. We have uploaded the search strategy as supplementary material. In order to obtain as much IPL related literature as possible, we mainly searched a variety of terms of IPL, and at the same time added the citation search of key literature. In this way, the relevant literature can be collected as comprehensively as possible.

Answer: We apologize for our oversight. We have rechecked the fund information, and the conclusion is that the fund name and number are correct.

4. Data Availability statement

Answer: Yes. We are not going to change the data availability statement. Thank you very much!

5. PLOS requires an ORCID iD for the corresponding author in Editorial Manager on papers submitted after December 6th, 2016.

Answer: We appreciate your suggestion. As the corresponding author, I applied for a new ORCID with ID 0000-0003-1948-2620. If you need more information, please contact us.

6. We note that you have provided funding information that is not currently declared in your Funding Statement. However, funding information should not appear in the Acknowledgments section or other areas of your manuscript.

Answer: We apologize for our non-standard format. We have removed this part from the revised manuscript. In the acknowledgments section, we express our gratitude to the professionals in literature screening and data extraction, and we express the non-relevance statement in the cover letter. Finally, thank you for your help in correcting our flaws in the online submission form.

7. Please include a separate caption for each figure in your manuscript.

Answer: We appreciate your suggestions. We have made modifications according to the requirements of the journal and hope to meet the requirements.

8. Please include your tables as part of your main manuscript and remove the individual files. Please note that supplementary tables (should remain/ be uploaded) as separate "supporting information" files.

Answer: We appreciate the editor's suggestion. We have uploaded the additional materials as supplementary materials.

9. Please review your reference list to ensure that it is complete and correct.

Answer: We appreciate the editor's suggestion. According to the requirements of the journal, we modified the format of the references and checked the information of the references, hoping to meet the requirements of the journal

Response to reviewer#1:

1. Why was this review not registered at Prospero ? 

Answer: We really appreciate your proposal. Without registration PROSPERO, we have two main considerations that we hope you can understand: First, our study relies on literature review to propose a new classification. Throughout the process, we focused on classification and treatment decision trees. Second, we used electronic and manual retrieval methods. The simple search strategy was designed to collect as much literature as possible on IPL, which may not be suited to PROSPERO's protocol. Therefore, we did not initially register with PROSPERO. But there is no denying that the evidence would be better if pre-registered with PROSPERO. We apologize for neglecting PROSPERO's registration by focusing on the design of the new classification and treatment decision tree. To complete this research, we have supplemented the PROSPERO registration (CRD42022378001).We hope it will not affect the quality and innovation of our study.

2. Proper analysis method was not adapted using PICO. 

Answer: We thank the reviewers for pointing out this problem. We initially focused on the histological characteristics of IPL and did not describe the PICO standards in detail, which may have led to the misunderstanding of the reviewers. We have now re-modified the PICO. However, the reports of IPL are cohort studies or case reports, and there is no control group, so we cannot describe the part of “C”.

3. Since antibiotics didn't play a strong role in resolution of the lesion, then why was it still advocated in the treatment decision tree ? 

Answer: This problem raised by the reviewer is caused by our insufficient expression. We sincerely apologize for it. In the discussion section, we mentioned that antibiotics could be tried, but did not point out the source of our opinion. In fact, our review of the literature found that a small number of scholars reported cases of IPL being cured with antibiotics alone (amoxicillin or amoxicillin and acetaminophen). However, their reports did not involve histopathology, which made it difficult to determine their category and therefore they did not meet our inclusion criteria. We have made a detailed elaboration of this situation in the paper, hoping that this will not affect readers' reading.

4. In case of the antibiotics usage, what was the dosage and type recommended ? 

Answer: We thank the reviewer for pointing out this part. Since many studies have failed to completely cure IPL by using antibiotics alone, some scholars have still achieved success. By reviewing the limited evidence, we found that they were inconsistent in dose and type, depending on the surgeon's experience and the characteristics of the IPL. There is no consensus or guidelines, but it is important to note that antibiotic use should be measured according to the cause, symptoms, or opening/closing of IPL. In addition, although antibiotics alone failed in some cases, reducing the discomfort of patients, and limiting the progression of the disease in stages was also mentioned as an advantage. Therefore, we add these views in the discussion section, hoping to be helpful to the paper (Page 21).

5. Were all the included patients healthy ?

We thank the reviewer for pointing out this aspect. Since most of the included studies were case reports, individual differences (age, sex, implant type) were difficult to regulate, which is different from the criteria for participants in cohort studies. In order not to affect the results of the study, we re-reviewed the included 28 articles and found that all participants did not have systemic diseases that induced IPL. In the interest of gathering the histological features of IPL as much as possible, we have not restricted the condition of participants (no systemic disease).

Response to reviewer#2:

1. In proposed classification and management, the Author did not take into account soft tissue changes. for example fig 2e could propose soft tissue challenges. 

Answer: We would like to thank the reviewer for his valuable questions on soft tissue management, which is a deficiency that does exist in our manuscript. In our classification, mucosal swelling and fistulas are described in Classification 2. Correspondingly, we did not cover the management of soft tissue in the discussion. We have to admit that the management of soft tissue is very important for peri-implant health. However, in none of the studies we included which the authors provide detailed evidence for managing soft tissue. This makes it impossible to describe IPL's soft tissue management protocols clearly. But as a rule of thumb, soft tissue abnormalities generally recover with the removal of the lesion, which may be a reason they were not described in the study. Based on the available evidence, we were careful in our discussion part to recommend the importance of soft tissue healing and need to maintain close attention (Page22. Adding: Besides, the healing of soft tissue involved in IPL are also of concern, especially in cases of mucosa fistula.). We hope this answer can be supported by the reviewers. 

2. It is a systematic review therefore statistical analysis is not applicable. 

Answer: We thank the reviewers for the suggestion. Since this study is a re-enumeration of previous studies, we did miss this point. Therefore, we modified "data synthesis" to "information synthesis", which is more consistent with the system evaluation. In addition, we have simplified the content for readers to understand.

3. Minor typing and grammatical errors. 

Answer: We apologize for some language and grammatical errors in our manuscript. We spent a lot of time on the manuscript, and additions and deletions may have contributed to the problem. We have now made further changes to the language and grammar, and have had a professional English scholar to proofread it. We hope this manuscript has been improved.

4. In result, you did not report the quality you assign to each study. Either summarised in the results section or as an extra column in your study characteristics table. 

Answer: We apologize for the irregular chart arrangement of the first draft which made it difficult for reviewers to read. NOS was used to assess the quality of the included case reports and cohort studies, and the results are listed in Table 3. I hope the layout of this time will not affect the reviewer's reading.

5. Most of the studies have one or two cases therefore any management recommendation should be advised cautiously. 

Answer: We sincerely thank the reviewers for their suggestions. We reviewed the discussion section of the manuscript to make it as referential and objective as possible.

6. Any recommendation on antibiotics protocols, such as before or after procedure, and what antibiotics to be considered?

Answer: We sincerely admit that we did not describe the use of antibiotics in detail, and thank the reviewer for pointing out this problem. In fact, in the discussion section, we briefly mentioned antibiotics and suggested trying them in Category 1. The reason for this is that we found several studies in the literature we reviewed (where no histological evidence was excluded) that completely cured IPL with antibiotics alone. Among them, they use different antibiotics and dosages, which makes the available evidence not a reasonable guide or recommendation. In this case, we added the key elements mentioned above to the discussion and recommended the use of antibiotics (e.g. Metronidazole, amoxicillin) according to the cause, symptoms and signs of IPL.

The above is my reply to editors and reviewers. If you have any questions, please contact me in time. Sincerely thank you for your suggestions to make our study better!

Best regards!

 Yours sincerely,

 Dr. Mei Mei

---

## [Editor Report · Decision Letter 1]

7 Dec 2022

A novel histopathological classification of implant periapical lesion: a systematic review and treatment decision tree

PONE-D-22-29419R1

Dear Dr.Mei Mei,

We’re pleased to inform you that your manuscript has been judged scientifically suitable for publication and will be formally accepted for publication once it meets all outstanding technical requirements.

Kind regards,

Fahad Umer

Academic Editor

PLOS ONE
---

## [Editor Report · Acceptance letter]

13 Dec 2022

PONE-D-22-29419R1 

A novel histopathological classification of implant periapical lesion: a systematic review and treatment decision tree 

Dear Dr. Mei:

I'm pleased to inform you that your manuscript has been deemed suitable for publication in PLOS ONE. Congratulations! Your manuscript is now with our production department. 

Kind regards, 

on behalf of

Dr. Fahad Umer 

Academic Editor

PLOS ONE